# A bionic stretchable nanogenerator for underwater sensing and energy harvesting

Yang Zou [1,2,3,7], Puchuan Tan[1,3,7], Bojing Shi[1,2,7], Han Ouyang [1,3], Dongjie Jiang[1,3], Zhuo Liu[1,2], Hu Li[1,2], Min Yu[4], Chan Wang[1,3], Xuecheng Qu[1,3], Luming Zhao[1,3], Yubo Fan[2,5], Zhong Lin Wang [1,3,6] & Zhou Li [1,3]

Soft wearable electronics for underwater applications are of interest, but depend on the development of a waterproof, long-term sustainable power source. In this work, we report a bionic stretchable nanogenerator for underwater energy harvesting that mimics the structure of ion channels on the cytomembrane of electrocyte in an electric eel. Combining the effects of triboelectrification caused by flowing liquid and principles of electrostatic induction, the bionic stretchable nanogenerator can harvest mechanical energy from human motion underwater and output an open-circuit voltage over 10 V. Underwater applications of a bionic stretchable nanogenerator have also been demonstrated, such as human body multi-position motion monitoring and an undersea rescue system. The advantages of excellent flexibility, stretchability, outstanding tensile fatigue resistance (over 50,000 times) and underwater performance make the bionic stretchable nanogenerator a promising sustainable power source for the soft wearable electronics used underwater.

[1] CAS Center for Excellence in Nanoscience, Beijing Key Laboratory of Micro-nano Energy and Sensor, Beijing Institute of Nanoenergy and Nanosystems, Chinese Academy of Sciences, Beijing 100083, China. [2] Beijing Advanced Innovation Centre for Biomedical Engineering, Key Laboratory for Biomechanics and Mechanobiology of Ministry of Education, School of Biological Science and Medical Engineering, Beihang University, Beijing 100083, China. [3] School of Nanoscience and Technology, University of Chinese Academy of Sciences, Beijing 100049, China. [4] School of Stomatology and Medicine, Foshan University, Foshan 528000, China. [5] National Research Center for Rehabilitation Technical Aids, Beijing 100176, China. [6] School of Materials Science and Engineering, Georgia Institute of Technology, Atlanta, Georgia 30332, United States. [7] These authors contributed equally: Yang Zou, Puchuan Tan, Bojing Shi. Correspondence and requests for materials should be addressed to Y.F. (email: yubofan@buaa.edu.cn) or to Z.L.W. (email: zhong.wang@mse.gatech.edu) or to Z.L. (email: zli@binn.cas.cn)

Next-generation wearable electronics are soft devices that are flexible, deformable, stretchable, biocompatible and waterproof[1–5]. Rapid development of wearable electronics is accompanied by a demand for underwater power conversion and supply. Significant progress in flexibility of capacitors[6], solar cells[7,8], biofuel cells[9,10] and nanogenerators[11–15] offers promise for powering wearable devices. Triboelectric nanogenerators (TENGs) have been invented as mechanical energy harvesters based on coupling effects of triboelectrification and electrostatic induction[16–19]. The advantages of outstanding flexibility, high energy conversion efficiency, light weight, low cost and simple fabrication make TENGs suitable as power supplies for electronics[20–23]. However, the difficulties in encapsulation, grounding and stability have largely limited the developments of TENGs for underwater applications[24,25].

Bionics is an important trend for technology development. Various bionic devices have emerged in an endless stream in recent years[26–28]. Electric eel (electrophorus electricus) can generate thousands volts of electricity under water due to its special electric organs[29]. Scientists have invented several electric generators based on ion-concentration gradients, which is a key mechanism of electric generation in electric eels[30–32]. Here we propose a bionic stretchable nanogenerator (BSNG) inspired by an electric eel. By mimicking the structure of ion channels on the cytomembrane of electrocyte in an electric eel, a mechanical control channel is manufactured by the effect of stress-mismatch between polydimethylsiloxane (PDMS) and silicone. Two kinds of unique working modes allow the BSNG to achieve over 170 V open-circuit voltage in dry conditions and over 10 V in liquid environment, which are combined with the advantages of the TENG and can be used for energy harvesting and underwater sensing. Owing to its advantages of excellent flexibility, stretchability, mechanical responsiveness and output performance, the BSNG is expected to be a human body motion monitor and a promising alternative power source for wearable electronics in dry and wet environments.

## Results

**Mechanism of bionic stretchable nanogenerator.** Electric eels can generate high voltages up to 600 V by stacking thousands of electrocytes in series (Fig. 1a)[33,34]. In the inactive state, compared with the external cell membrane, the resting electrocytes contain higher concentration of potassium ions (K[+]) and lower concentration of sodium ions (Na[+]). In the active state, the structure of ion channels on cell membranes is triggered by neurotransmitter, which allows the ions to pass through the cell membranes driven by a polarized concentration gradient of Na[+] and K[+] (Fig. 1b). The transmembrane potential of individual electrocyte can be raised to ~150 mV[34,35]. Therefore, a large amount of electrocytes stacked can generate electricity with very high voltage (Fig. 1c).

The BSNG is inspired by the electricity generating principle of electric eel. The BSNG has two layers (Fig. 1d). The first layer is electrification layer that contains a series of controllable channels based on PDMS-silicone double layer structure, and a fluid chamber filled with electrification liquid (Fig. 1e). The electrification liquid used in BSNG is deionized (DI) water, with the advantages of high dielectric constant, good bio-safety, low cost and easy preparation. When combined layer by layer after curing, the PDMS and silicone form a strong binding force in the junction, which leads to a bend of composite towards PDMS layer (Supplementary Figure 1, Supplementary Note 1). The interface of two materials is well fused. The similar infrared spectrum results show that PDMS, silicone and the composite have very similar molecular groups (Supplementary Figure 2).

The PDMS layer is cut into multiple sections, thus the PDMS sections will be separate from each other after stretching. Then, the separated PDMS sections will retract for the resilience of silicone when releasing (Supplementary Figure 1). Therefore, multiple channels can be controlled by tension to open or close simultaneously (Supplementary Figure 3, Supplementary Note 2). To make the channels more hydrophobic for liquid flowing, some bamboo joint-liked microstructures at the bottom of the channels and reservoir are built by reversing 3D printed mold (Supplementary Figures 3, 4, Supplementary Note 3). The second layer is induction layer, which contains two ionic solution electrodes under the channels and chamber of the first layer. The ionic solution used in BSNG is sodium chloride (NaCl) solution. The whole dimensions of the BSNG is 10 cm × 6 cm × 8 mm. The detailed size of each part structure of BSNG can be found in Supplementary Figure 5 and Supplementary Note 4.

The channels of the BSNG mimic the ion channels embedded in cell membrane of electrocytes, which are constituted by special proteins. The activity of ion channels regulates the ions in and out by opening and closing ion channels, which is vital to realize various functions of cells. The Na[+] and K[+] channels of electrocytes in electric eel are gated by voltage and neurotransmitter, which play a crucial role in electricity generation. More details about the Na[+] and K[+] transfer are showed in supplementary materials (Supplementary Figure 6, Supplementary Note 5). The BSNG mainly uses the triboelectrification caused by liquid flowing and electrostatic induction principle to generate electricity, which is different from previous reported work about generator harnessed ion-concentration gradients[30–32]. In one working cycle, the BSNG can produce an open-circuit voltage up to 10 V underwater (Fig. 1f).

To achieve artificial control of the flowing of electrification liquid, we propose the mechanosensitive channel inspired by the mechanically gated channel in cell membrane of electric eel. The multiple channels in the BSNG can open and close simultaneously controlled by a simple mechanical force. When applied a stretching force, the mechanosensitive channels in electrification layer of the BSNG are opened, and the liquid in the reservoir flow into the chambers spontaneously due to the negative pressure of the chambers. When the stretching stress is released, the separated chambers will be completely closed for the elastic resilience of silicone, resulting in that the liquid in chambers flows back into the reservoir (Fig. 2a, b). Red ink is injected into the liquid reservoir to see clear flowing process of the liquid.

The operating principle of the BSNG is based on the coupling of contact electrification and electrostatic induction. The conduction mechanism for electricity transport is attributed to the Maxwell's displacement current[36], which can be defined as:

$$J_D = \frac{\partial D}{\partial t} = \varepsilon_0 \frac{\partial E}{\partial t} + \frac{\partial P}{\partial t}, \qquad (1)$$

where $J_D$ is the free electric current density; $D$ is displacement field; $\varepsilon_0$ is permittivity in vacuum; $E$ is the electric field and $P$ is polarization field. The one working cycle of the BSNG is demonstrated in Fig. 2c. At initial state, the mechanosensitive channels in upper layer (electrification layer) are closed. Electrification liquid can only exist in a fixed reservoir lead to no potential change inside the BSNG. When the BSNG is stimulated by a mechanical traction, the channels open into a plurality of chambers. Electrification liquid flows into the chambers due to the negative pressure in the vacuum chambers, inducing a triboelectrification of flowing liquid. When the liquid contacts with the silicone substrate of chambers, negative ions are selectively absorbed onto the surface of the silicone substrate,

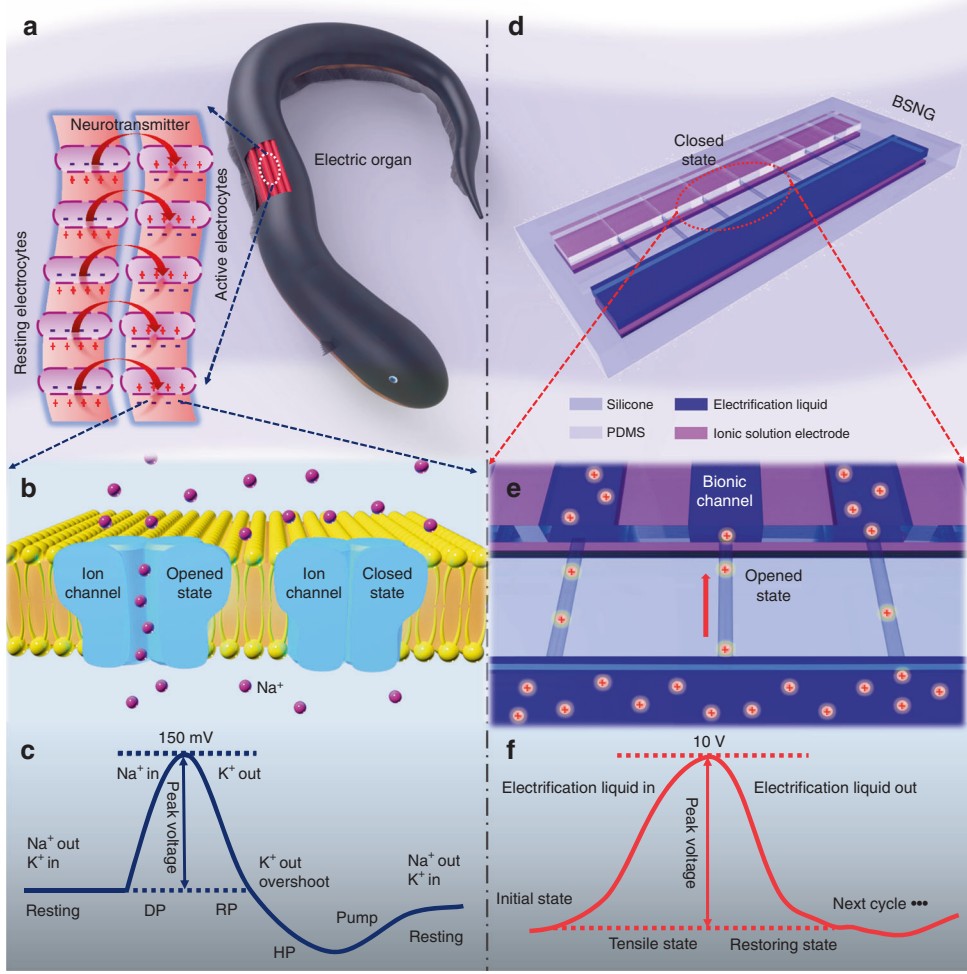

**Fig. 1** Bionic principle and structure of bionic stretchable nanogenerator. **a** Schematic diagram of electric eel and electrocytes. **b** Schematic diagram of ion channels on cytomembrane. **c** Action potential signal of electrocyte. DP, RP, and HP represent depolarization, repolarization, and hyperpolarization, respectively. **d** Scheme diagram of bionic stretchable nanogenerator (BSNG) with double layer structure, which is mainly constructed by silicone, polydimethylsiloxane (PDMS), electrification liquid and ionic solution electrode. **e** Scheme diagram of the bionic channels in BSNG. **f** Output signal of BSNG in one working cycle

resulting in an accumulation of negative triboelectric charges. The liquid in the BSNG is positively charged and the silicone near to the liquid is negatively charged by triboelectrification through the iterative liquid-silicone contact. The exact mechanism of the liquid-solid contact electrification is investigated by previously reported works[37–39]. In the meantime, the lower layer (induction layer) of the BSNG induces the accumulated electric charges on the bottom of upper layer due to electrostatic induction. Then the asymmetrical electric charges between the two liquid electrodes form a potential difference, which drives electrons to flow from one electrode to the other through external circuit, causing a current signal. When the mechanical traction exerted on the BSNG is removed, the chambers are closed due to the intrinsic resilience of BSNG. The electrification liquid in chambers is squeezed back into reservoir and leaving the silicone substrate of chambers. The absorbed ions remain on the surface of silicone and the excessive conjugate ions are form in liquid, the positive charges in the liquid asymmetrically screens the negative triboelectric charges on the surface of silicone, causing a reversed current in the external circuit. By applying and releasing a mechanical traction repeatedly, the back and forth movement of the electrification liquid induces a

continuous alternating electric signal between two ionic solution electrodes, and a continuous alternating current is generated through external circuit.

**Characterization of bionic stretchable nanogenerator.** The proposed principle of the working mode of the BSNG is confirmed through a finite element simulation using the COMSOL software. The potential distribution of the surface of the BSNG at a 50% strain state in cross section is shown in Fig. 2d. The detail information about the changes of the surface potential of the BSNG in different strain can be found in Supplementary Figure 7. Owning to the excellent stretchability of silicone and the segmented structure of PDMS, the BSNG can be stretched over 60% (Fig. 2e). Uniaxial tensile tests are performed to evaluate the mechanical properties of the BSNG, PDMS, silicone, PDMS-silicone double layer composite and BSNG (Fig. 2f). The BSNG can maintain its robustness when the tensile strain reaches 60%.

The BSNG is found to have outstanding performance in liquid environment. The underwater outputs of the BSNG stretched by hand are recorded in Fig. 3a–c. The peak open-circuit

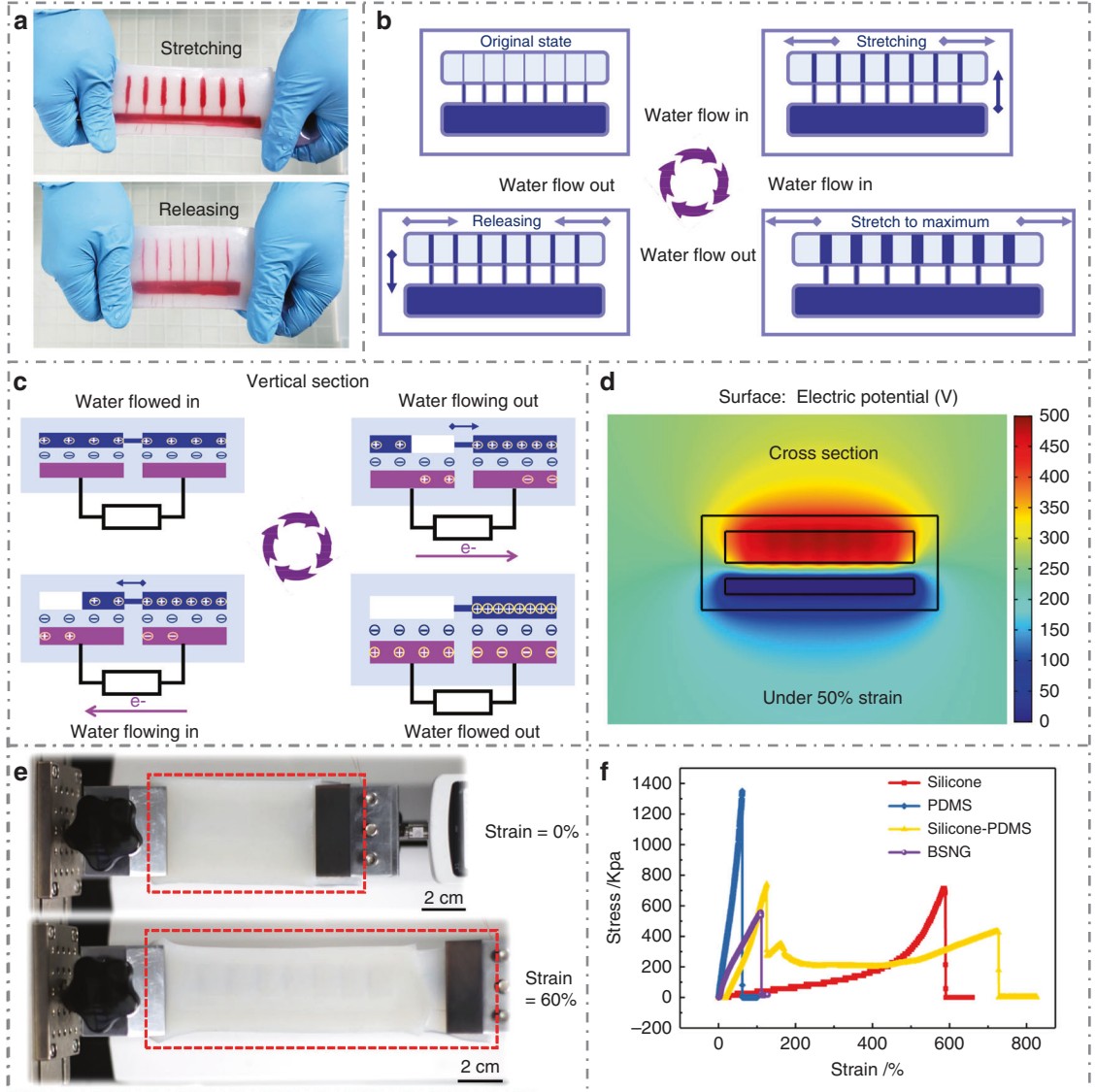

**Fig. 2** Working principle and stretchability of bionic stretchable nanogenerator. **a** Photographs of one working cycle of bionic stretchable nanogenerator (BSNG) (red ink filled in). **b** Schematic diagram of the working process of BSNG. **c** Schematic diagram of the working mechanism of BSNG. **d** Simulation result of BSNG under 50% strain. **e** BSNG (indicated by red frame) at initial state (0% strain) and stretched state (60% strain). **f** Uniaxial tensile test of the silicone, polydimethylsiloxane (PDMS), silicone-PDMS and BSNG

voltage ($V_{oc}$), short-circuit current ($I_{sc}$) and short-circuit transferred charge ($Q_{sc}$) can reach 10 V, 36.5 nA and 2 nC, respectively (1 Hz tensile frequency, 50 % tensile strain). The relationship between the output and the different strain state of the BSNG are also explored (Fig. 3d–f). The results show that the $V_{oc}$, $I_{sc}$ and $Q_{sc}$ of the BSNG increase nearly one fold as the strain increases from 10 to 50%, respectively. This is due to the output of TENG is positively correlated with the actual area of contact separation between the two materials[36]. In this work, the output of BSNG is directly related to the size of the opened channels under different tensile strain correspondingly. Because the first 20% strain has little influence in channel's width, the difference of the output between 10% strain and 20% strain is not obvious.

Tensile frequency is one of important influence factors for the output performances of the BSNG. When the BSNG is stretched by a linear motor at different frequency from 0.5 Hz to 2.5 Hz, the $V_{oc}$ and $Q_{sc}$ increase slightly. Meanwhile, the $I_{sc}$ is nearly doubled from 50 nA at 0.5 Hz to 100 nA at 2.5 Hz

(Fig. 3g–i). The strain state of the BSNG is maintained at 50%. This is due to the $I_{sc}$ increased in parallel with the velocity. Based on a capacitor model, the output current of a BSNG can be defined as:

$$I = \frac{\partial Q}{\partial t} = C\frac{\partial V}{\partial t} + V\frac{\partial C}{\partial t}, \qquad (2)$$

where $Q$ is the stored charges in the capacitor, $C$ is the capacitance of the BSNG and $V$ is the voltage between the two electrodes[36].

The silicone layer around BSNG acts as an insulating layer just like the thick adipose layer of the electric eel, which protects the charge transfer process in the BSNG from being influenced by electrostatic shielding of water significantly. It is the reason why the BSNG still exhibits a good output performance in liquid environment. It is noted that the underwater output of BSNG is attenuated compared to the test results out of the water. This is mainly due to the electrostatic shielding of water existing around the BSNG, the charges in the water

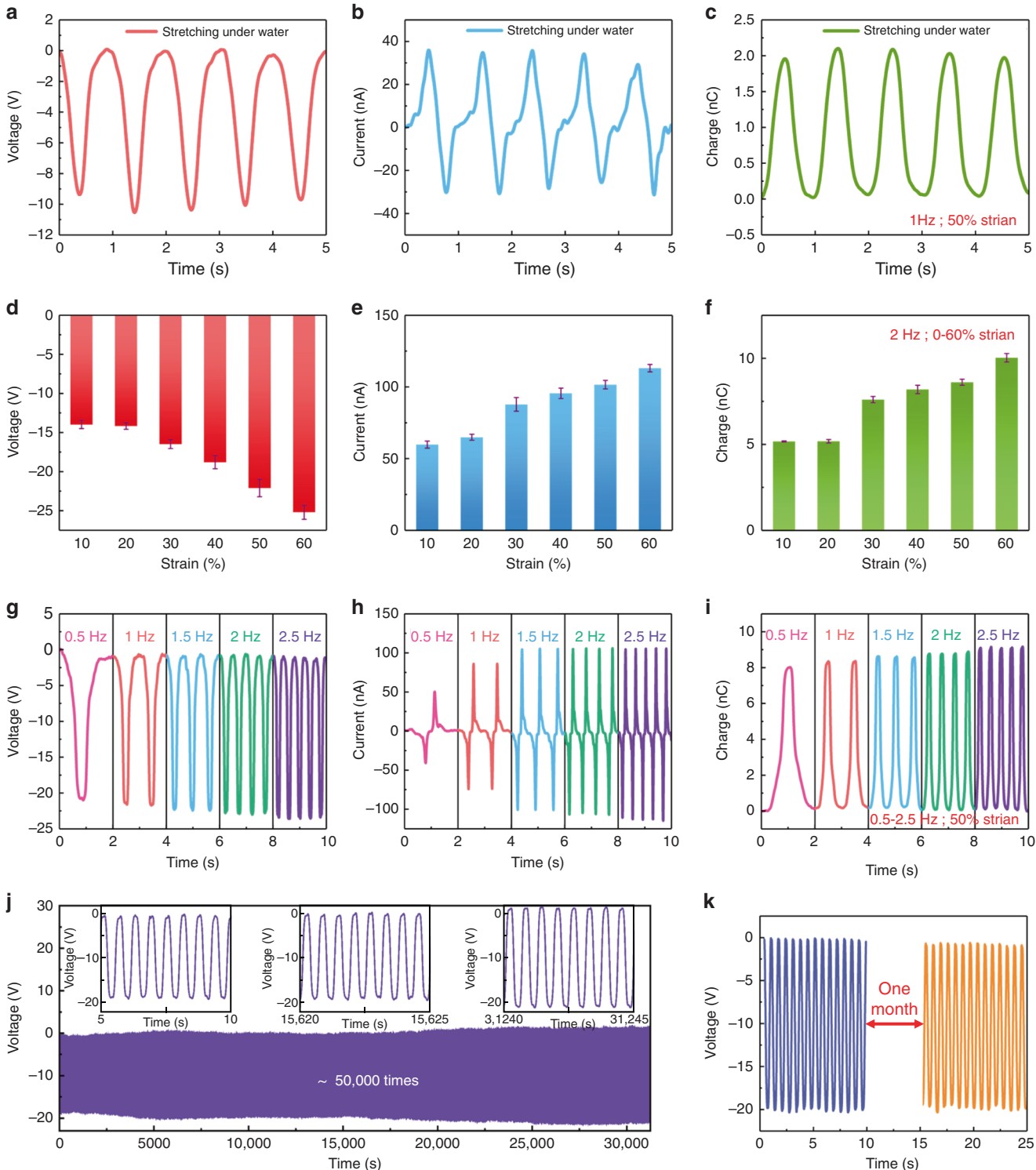

**Fig. 3** Electrical characteristics of bionic stretchable nanogenerator. **a** Open-circuit voltage $V_{oc}$, **b** Short-circuit current $I_{sc}$, and (**c**) short-circuit charge quantity $Q_{sc}$ of BSNG when working underwater (stretched by hand, under 50% strain, at 1 Hz). **d** $V_{oc}$, (**e**) $I_{sc}$, and (**f**) $Q_{sc}$ of BSNG when working under different strain (stretched by a linear motor, at 2 Hz). **g** $V_{oc}$, (**h**) $I_{sc}$, and (**i**) $Q_{sc}$ of BSNG when working at different frequency (stretched by a linear motor, under 50 % strain). **j** $V_{oc}$ of BSNG that lasted for ~50000 cycles stretched by a linear motor (31250 s, under 50 % strain, at 1.6 Hz). The details within 5 s at the beginning, in the middle part and at the end of test are shown in the top small figures, respectively. **k** Comparison of the $V_{oc}$ of BSNG before and after placed in normal temperature environment for one month. Source data of (**d**–**f**) are provided as a Source Data file. All data in (**d**–**f**) are presented as mean ± s.d.

neutralize partial charges on the surface and inside of the BSNG. It is worth mentioning that the BSNG can also work in a different mode similar to single-electrode TENG, which can generate a peak $V_{oc}$ over 170 V (Supplementary Figure 8, Supplementary Note 6) under a dry conditions but no obvious output under a wet conditions (Supplementary Figure 9, Supplementary Note 7). Since there have been many reports on single-electrode TENG before, we will not repeat here. Detailed working principle (Supplementary Figure 10, Supplementary Note 8) and related output performance characterization results of BSNG working in single-electrode mode can be found in supplementary materials. The power density curves of BSNG in single-electrode mode and liquid-solid contact mode are shown in Supplementary Figure 11, respectively. In single-electrode mode, the peak power density of BSNG can reach 18 mW m$^{-2}$ when the external load is 50 MΩ. In liquid-solid contact mode, the peak power density of BSNG can reach 62.5 µW m$^{-2}$ when the external load is 300 MΩ.

Tensile fatigue resistance is an important property of stretchable devices. Here a 50 thousand times uniaxial tensile test is performed by using a linear motor to stretch BSNG to 50 % strain repeatedly at a frequency of 1.6 Hz. The whole process lasts 31250 s. The corresponding output $V_{oc}$ of the BSNG in overall process is recorded in Fig. 3j. The tensile fatigue test shows that the BSNG has outstanding durability and tensile fatigue resistance. Even after about 25 thousand working cycles, the $V_{oc}$ of the BSNG maintains stable compared with its initial state ($V_{oc}$ = ~20 V). When it reaches about 50 thousand working cycles, the $V_{oc}$ of the BSNG is slight increase. The possible explanation could be that certain charges were accumulated in the channels inside the BSNG after a certain number of friction cycles[40,41]. The characterization of the internal structure of BSNG after 50 thousand times fatigue test demonstrates that the bionic channels inside BSNG is intact (Supplementary Figure 12, Supplementary Note 9). Benefiting from the small friction stress between liquid and soft matter, the bionic channels inside BSNG is not easily damaged even after a long period of reciprocating flow of the electrification liquid. Due to the good air tightness of silicone, after storing BSNG in the dry environment for one month, the output performance of BSNG shows no obvious degradation (Fig. 3k).

**Underwater sensing and energy harvesting**. The BSNG can be used for scavenging mechanical energy and monitoring the movement of human body in liquid environments. Excellent flexibility and stretchability make the BSNG suitable for wearing and adapt for human body. Taking full advantage of flexibility, stretchability and good mechanical responsiveness in liquid environment of the BSNG, a set of human body multi-position motion monitoring and wireless transmission system used under water is constructed by integrating the BSNG and a packaged multi-channels wireless signal transmission module (Fig. 4a). A linear relationship is found between the curvature of the elbow and the output voltage of the BSNG fixed onto a human's arm to monitor the human motion (Fig. 4b). An integrated wearable BSNG is developed for the system through fixing the BSNG onto a silicone wristband. A total of four BSNGs are worn on the elbows and knees of human body respectively (Fig. 4c). The motion signals of four arthrosis can be acquired in real time through assorted software installed in a laptop (Supplementary Movie 1). The system is tested in a conventional swimming pool. A volunteer wears the BSNG based system and swims in different swimming styles. When diving into the water, the motion signals of different parts of human body is displaying and recorded on the computer screen in real

time. The results show that the amplitude of motion signal is maximum when breaststroke, owning to the large amplitude movement of arm swing and leg driving (Fig. 4d). The frequency of motion signal is the highest while swimming freestyle, and the lowest during backstroke. When treading water, only motion signals of leg are acquired while that of arm is not obvious. The data of the recorded motion signals of each swimming stroke have been further analyzed in Supplementary Figure 13 and Supplementary Note 10. The average motion amplitude and average time interval of the peaks of the motion signals of each stroke have been extracted, respectively. With the information acquired from BSNGs, we can analyze the specific case of each movement of the swimmer to estimate the physiology states under the water. The system can be used for supervising movements and training of swimmers. A simulated drowning signals of human body can also be recorded by the system (Supplementary Figure 14). The wireless motion monitor system based on the BSNG is proved to be a role of "black box" for swimmers being in danger.

Undersea rescue is another important application for the BSNG. An undersea rescue system is constructed by the BSNG, wireless transceiver module and a warning light (Fig. 5a–c). The simple circuit diagram of the system is shown in Fig. 5d. The four wearable BSNGs integrated with a capacitor are used to harvest the mechanical energy from the human motions to be used as a power source for radio remote controller. A 100 µF capacitor is connected to the four BSNGs in parallel through rectifier respectively to test the ability of BSNG charging for an energy storage device. The voltage of the capacitor is charged from 0 to 3 V in about four and a half hours, which can drive wireless transmitter to emit a trigger signal one time (Fig. 5e, Supplementary Figure 15, Supplementary Note 11). Then the 100 µF capacitor charged by four rectified BSNGs wore on human body for about 4 h, a wireless transmitting signal can be triggered to remotely switch a red warning light (Fig. 5f, Supplementary Movie 2). In addition, when tapping the BSNG on the shoulder of human body, a SOS rescue light composed by 12 red LEDs can be easily lighted up (Supplementary Figure 14, Supplementary Movie 3). The excellent underwater working ability makes the BSNG play an important role in underwater applications, acting as an energy harvesting device or a self-powered sensor. It is noted that BSNG also has a wealth of applications in daily life (Supplementary Figure 16, Supplementary Note 12), such as harvesting energy or self-powered monitoring when body-building, establishing human-machine interface, and so on.

**Discussion**
We have proposed a stretchable bionic-electrocyte nanogenerator based on liquid- electrification for underwater energy harvesting and sensing applications. Different from the previously reported bionic electric eel generators utilizing the energy from the ion-concentration gradients, the presented BSNG mimics the structure of ion channels on cytomembrane of electrocyte to realize a circular flowing of electrification liquid in BSNG by mechanical stress, which results in a continuous alternating electric current. The mechanosensitive channels in the BSNG are fabricated by a PDMS-silicone double layer structure, utilizing the effect of stress mismatch between the two materials. The combinations of soft materials including silicone, PDMS, liquid electrode and electrification liquid, endow the BSNG with good flexibility, stretchability and outstanding tensile fatigue resistance. In addition, it is also possible to design the BSNG into arbitrary and complicated shapes by changing different structures of the channel. These characters are essentials for soft wearable

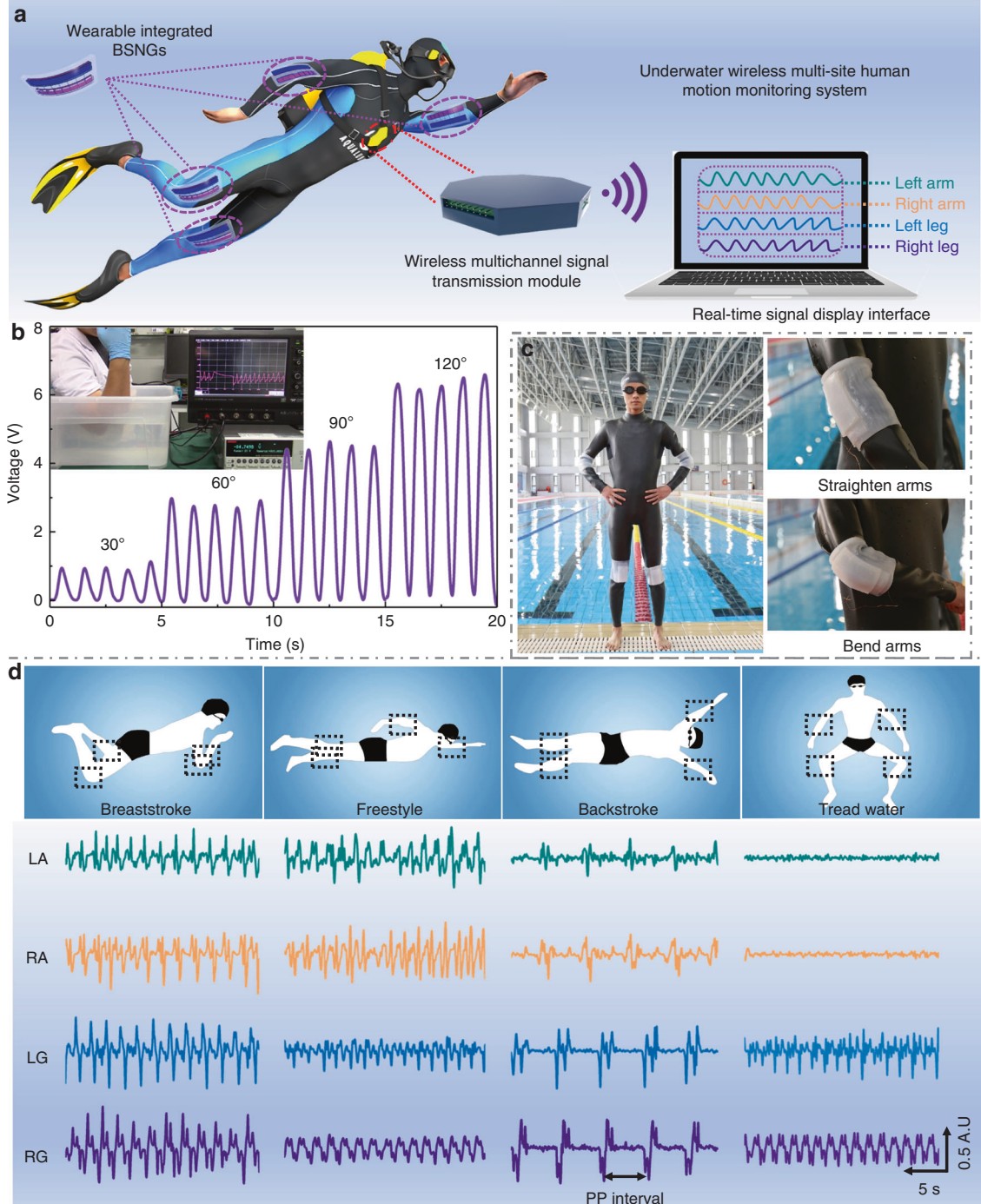

**Fig. 4** Underwater wireless multi-site human motion monitoring system. **a** Illustration of underwater wireless multi-site human motion monitoring system based on bionic stretchable nanogenerator (BSNG). **b** Signal outputs of BSNG fixed on the elbow at different curvature motion. **c** Photographs of integrated wearable BSNG worn on the arthrosis of human. **d** Signal outputs recorded by underwater wireless multi-site human motion monitoring system when the volunteer swam in different strokes (LA, RA, LG, RG represent left arm, right arm, left leg, right leg, respectively; PP interval represents time interval between two peaks)

electronic devices[42–46]. Good performance in liquid environments makes BSNG a potential power supply for wearable device used in moist environment even underwater. The excellent mechanical responsiveness further leads to the BSNG as a motion monitor for swimmer and other underwater workers.

Even though the BSNG in this work is flexible, stretchable and behaving well underwater, there is still space for improvement.

The dimensions of BSNG could be smaller and thinner in virtue of developments of soft fabrication techniques in the future. A miniaturized BSNG has a potential to act as a body mechanical energy harvester or sensor for implantable applications, such as harvesting heart beating energy and sensing pulse signals[47–51]. In addition, some advances of material and surface modification for maximizing surface electrostatic charge density while liquid flowing, together with the new packaging strategy for reducing

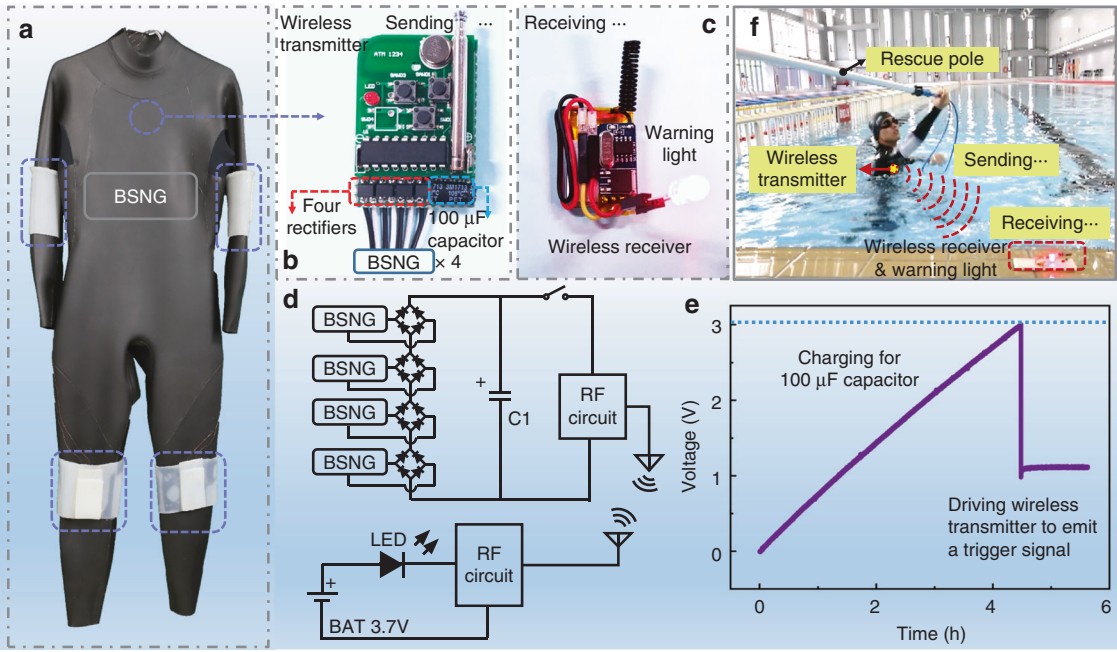

**Fig. 5** Undersea rescue system based on bionic stretchable nanogenerator. Photograph of undersea rescue system which included (**a**) integrated energy harvesting diving suit, (**b**) integrated wireless transmitter and (**c**) wireless receiver integrated with a red warning light. **d** Simple circuit diagram of undersea rescue system. **e** Voltage changes of a 100 μF capacitor charged by BSNG and used to power a wireless transmitter to emit a trigger signal. **f** Physical map of undersea rescue system sending an alert when swimmer in danger (red LED was lighted up remotely)

electrostatic shielding underwater could be used to further enhance the performance of BSNG. With these improvements, BSNG could have more opportunity for potential applications in electrical skins, soft robots, wearable electronics and implantable medical devices.

## Methods

**Materials**. The silicone was used EcoflexTM 00–30 produced by Smooth-On, Inc. The PDMS was from Dow corning. The NaCl was from Sigma-Aldrich. A 3D printer (Raize 3D) and polylactic acid (PLA) printing-supplies (Raize 3D) were used for designing and printing molds of channels and reservoirs inside BSNG. Several customized Polytetrafluoroethylene (PTFE) molds were used as containers to cure components of BSNG. A commercial OpenBCI 8 bit board (3IT_EEG OBCI Kits) was used for wireless gathering and transmitting data. A commercial radio remote controller was used to fabricate undersea rescue system. A commercial diving suite was used for swimming test under water. Red commercial light-emitting diodes (LEDs) were used as warning lights for rescue.

**Fabrication of polydimethylsiloxane-silicone double layer composite**. The PDMS-silicone double layer composite was fabricated by two-step method. First, a layer of silicone was prepared. Required amounts of Parts A and B of EcoflexTM 00–30 were dispensed into a mixing container (1A: 1B by volume or weight) and were mixed thoroughly for 3 min. Then the mixture was poured in another container for curing. Vacuum degassing was required to eliminate any entrapped air of mixture. After vacuumized for 3–5 min, the mixture within container was curing at 50 °C for 2 h. The second step was curing a layer of PDMS onto the cured silicone layer. The PDMS main agents and curing agents (10:1 by weight) were mixed thoroughly. Then the mixture was vacuumized for 30 min before pouring into the container with cured silicone prepared before. PDMS was required to completely and evenly cover on the surface of the silicone. After curing at 80 °C for 4 h, a PDMS-silicone double layer composite was completed.

**Fabrication of bionic stretchable nanogenerator**. The Supplementary Fig. 16 showed the fabrication process of the BSNG. First, the master molds of channels and reservoirs were fabricated by 3D printing PLA supplies. Then the master molds were fixed onto the PTFE mold by Kapton tape. The prepared gel of silicone (Ecoflex 00–30) was poured into the PTFE mold and cured at 50 °C for 2 h. After that, the formed silicone components were removed, which mainly included encapsulation layer, induction layer, and electrification layer from top to bottom. The preformed gel of PDMS was poured into the groove of electrification layer, curing at 80 °C for 4 h. Then, the PDMS layer was cut into 8 sections

and constructed 7 channels. Two varnished wires were sealed in reservoirs of induction layer by preformed gel of silicone, curing at 50 °C for 2 h. Then 3 components were assembled into a whole device with preformed gel of silicone and PDMS, curing at 80 °C for 4 h. At last, electrification liquid such as DI water was injected into the reservoir of electrification layer and sodium chloride solution (1 M) was injected into the reservoirs of induction layer as electrodes.

**Fabrication of wireless motion monitoring system**. A wireless motion monitoring system was constructed by four integrated wearable BSNGs (BSNG fixed onto a silicone wristband) and a multi-channels wireless signal transmission module (including an OpenBCI 8 bit Board and an OpenBCI programmable dongle). The four integrated wearable BSNGs wore on the elbows and knees of human body were connected with an OpenBCI 8 bit Board through varnished wires directly to gather and transmit motion signals from different body parts. The OpenBCI programmable dongle was connected with a laptop via USB port to receive signals. An open-source software of OpenBCI is used to display motion signals in real time and record corresponding data.

**Fabrication of undersea rescue system**. An undersea rescue system was constructed by wireless launch parts and wireless receive parts. The four integrated wearable BSNGs fixed on a commercial diving suite was connected with a 100 μF capacitor through four rectifier bridges respectively. The capacitor was connected with the wireless launch module as a power source. All launch parts of the radio remote controller were fixed inside the diving suite. A red LED was connected with the wireless receive module as a warning light for rescue, and a 3.7 V–300 mAh lithium battery was connected to the wireless receive module for power supply.

**Characterization and measurement**. An ESM301/Mark-10 system was used for the mechanical tensile test and a Mark-10 force gauge was used to detect the applied force. The strain rate was fixed at 120 mm/min for the tensile test. A linear motor (LinMot E1100) was used to apply a periodic mechanical traction on BSNG continuously for maintaining working cycle. Different step speed and step distance of linear motor were set to control BSNG working under different frequency and different strain. The open-circuit voltage, short-circuit current and short-circuit charge of BSNG were measured by a Keithley 6517 electrometer, and the data were collected and recorded by an oscilloscope (LeCroy HDO6104). The scanning electron microscope (SEM) images were taken by Hitachi field emission SEM (SU 8020). The underwater data of swimmer were recorded by a wireless motion monitoring system based on an OpenBCI 8 bit board.

## Data availability

All data needed to evaluate the conclusions in the paper are present in the paper and/or the Supplementary Materials. The source data underlying Fig. 3d–f and Supplementary Figs 13a-l are provided as a Source Data file (https://doi.org/10.6084/m9.figshare.7892423.v1). Additional data related to this paper may be requested from the authors.

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

## Acknowledgements

We thank the University of Chinese Academy of Sciences for supporting the swimming venue for experiments. We are grateful to all the laboratory members for their cooperation in this study. This study was supported by the National Key R&D project from Minister of Science and Technology, China (2016YFA0202703), National Natural Science Foundation of China (No. 61875015, 31571006, 81601629 and 21801019), the 111 Project (Project No.: B13003), the Beijing Natural Science Foundation (2182091), and the National Youth Talent Support Program.

## Author contributions

Y.Z., P.T. and B.S. conceived the idea and designed the experiment. Z.Li, Y.B.F. and Z.L.W. guided the project. Y.Z. and P.T. designed and fabricated the BSNG. H.L. and C.W. performed the material characterization. B.S., Y.Z. and H.O. carried out the related electrical characterization. H.O. and Y.Z. designed the wireless motion monitoring system and the undersea rescue system. Y.Z., P.T., B.S., Z.Liu, M.Y., H.O., L.Z. and X.Q. carried out the underwater experiment and analyzed the experimental data. D.J., Y.Z. and P.T. drew the figures. Y.Z., B.S. and Z.Li prepared the manuscript. All authors discussed and reviewed the manuscript.

## Additional information

**Competing interests:** The authors declare no competing interests.

