## [Peer Review File · Nature Communications]

Reviewers' comments:

Reviewer #1 (Remarks to the Author):

In this paper, authors suggest a new soft wearable electronics for underwater applications which provides a waterproof, long-term sustainable power source. Also, various experiments were well conducted to optimize properties of BSNG. I thought that your idea can expand filed of application of triboelectric nanogenerator. I read your paper carefully and I have some questions and suggestions.

1. In figure 1c, what is the peak voltage value of 150mV? Is this actually measured?
2. In figure 1d, what is the kind of electrification liquid? You must mention about the specific characteristic of the material, you used.
3. In figure 2c, it is necessary to express the display of the charge in a little more detail. The number of electrons when the electrification liquid enters the bionic channel and the amount of charge in the ionic solution electrode are not proportional to the figure.
4. In figure 3d, is there any reason to represent the voltage value as a negative value? Looking at figures e and f, short-circuit current and short-circuit charge are expressed as positive values, while open-circuit voltage is expressed as negative values.
5. In figure 4d, we think there will be more important factors that can show differences in each posture. For example, the time difference between each cycle needs to be analyzed.
6. In figure 5, regarding the Undersea rescue system, you said that the wireless transmission and reception were done by charging the capacitor for 4 hours and 30 minutes. I wonder that the person must swim without resting for 4 hours and 30 minutes to charge the capacitor, and I wonder how the connection between wireless transmitter and the rectifier is made. We think you can improve applications using BSNG, is there other examples that can expand application of BSNG?
7. In Supplementary Figure 1a, it needs more information about "CTE"
8. In figure 2e, it needs to be uniformity of spacing, "Strain =0% > Strain=0%"
9. In figure 5e, the expression "simple circuit diagram" is inappropriate.
10. In figure 5f, what is the white pole that the subject's right hand is holding?

Reviewer #2 (Remarks to the Author):

In this paper, authors introduced an interesting bionic stretchable nanogenerator (BSNG) inspired by the structure of ion channels on the cytomembrane of electrocyte. The device is with two different electricity generation modes, great stretchability and superior output performance under water, which is of great value to wearable electronic devices for underwater application. The human body multi-position motion monitoring and undersea rescue system based on the BSNG was demonstrated in this manuscript. They showed an outstanding tensile fatigue resistance over 50,000 times for the good opportunity as a soft, wearable, sustainable power sources used for underwater electronics. It is definitely a novel device of its unique design and interdisciplinary of nanogenerator, bionics and biomedical engineering, which is first presented in research works. The paper merits publication after a few minor changes.

Comment 1:

The macro size of the entire BSNG is actually not small, why is it still called a nanogenerator?

Comment 2:

Why do authors choose deionized water as the electrification liquid in BSNG instead of other liquids?

What are the advantages of deionized water compared to other liquids?

Comment 3:

The conductivity of sodium chloride solution is not excellent compared to other commonly used conductive ion electrode such as lithium chloride solution. Does the conductivity of the electrode influence the performance of the nanogenerator based device? Why to choose sodium chloride solution here?

Comment 4:

How to prove that bamboo joint-liked microstructures at the bottom of the channels can make the channels more hydrophobic? Please add a pro-hydrophobic test of the channels with and without bamboo joint-liked microstructures.

Comment 5:

What is the thickness of the electrification layer and induction layer of BSNG? Please specify the size of each part structure of BSNG in the manuscript, these data are important for this device.

Comment 6:

For a stretchable device, 50 thousand times of fatigue test results of BSNG are excellent. However, after 50 thousand times uniaxial tensile test, will the internal structure of BSNG has some changes? The reviewer suggests to add more details (e.g. photograph) to show the conditions of the internal structure of BSNG after uniaxial tensile test.

Comment 7:

The power density curve is also an important criterion for the BSNG. Please add relevant experimental data.

Comment 8:

What are the challenges of BSNG, and what improvements can be made afterwards? The authors should discuss more in the "Discussion" section.

Comment 9:

There are some basic language issues in the manuscript, the authors should check carefully and modify them.

Our point-by-point responses to reviewers' comments are detailed as follows. Responses are in blue. And the detailed revisions on our manuscript are highlighted.

Reviewer #1 (Remarks to the Author):

In this paper, authors suggest a new soft wearable electronics for underwater applications which provides a waterproof, long-term sustainable power source. Also, various experiments were well conducted to optimize properties of BSNG. I thought that your idea can expand field of application of triboelectric nanogenerator. I read your paper carefully and I have some questions and suggestions.

Comment 1:

In figure 1c, what is the peak voltage value of 150mV? Is this actually measured?

Responses:

Thank you for your time and attention to our manuscript. Here, the peak voltage value of 150 mV is a transcellular potential difference of an electrocyte during a nerve impulse, which is not an actually date we measured. We got relevant information of this voltage value from the following three references:

[1] An electric-eel-inspired soft power source from stacked hydrogels[J]. *Nature*, 2017, 552(7684): 214.

[2] Electrophorus electricus as a model system for the study of membrane excitability[J]. *Comparative Biochemistry and Physiology Part A: Molecular & Integrative Physiology*, 1998, 119(1): 225-241.

[3] High-voltage nanofluidic energy generator based on ion-concentration-gradients mimicking electric eels[J]. *Nano Energy*, 2018, 43: 291-299.

Figure R1. Schematic diagram of electrocytes in resting and stimulated state. (Reference: Electrophorus electricus as a model system for the study of membrane excitability[J]. Comparative Biochemistry and Physiology Part A: Molecular & Integrative Physiology, 1998, 119(1): 225-241.)

The generation process of the transcellular potential can be concluded as follows:

At rest state, both the innervated and noninnervated membrane exhibit a potential of -85 mV. When stimulated, activated acetylcholine receptors (AchRs) generate endplate potentials, triggering Na⁺ channel-mediated action potentials peaking at +65 mV on the innervated membrane. The noninnervated membrane contains no voltage-gated Na⁺ channels and maintains resting potential at -85 mV. The result is the transcellular potential difference of an approximate value of 150 mV.

Revise in supplementary information:

(Page:10. Line:139-145)

The generation process of the transcellular potential can be concluded as follows: At rest state, both the innervated and noninnervated membrane exhibit a potential of -85 mV. When stimulated, activated acetylcholine receptors (AchRs) generate endplate

potentials, triggering Na⁺ channel-mediated action potentials peaking at +65 mV on the innervated membrane. The noninnervated membrane contains no voltage-gated Na⁺ channels and maintains resting potential at -85 mV. The result is the transcellular potential difference of an approximate value of 150 mV.

Comment 2:

In figure 1d, what is the kind of electrification liquid? You must mention about the specific characteristic of the material, you used.

Responses:

Thanks for your suggestion. The electrification liquid used in BSNG is deionized water. The deionized water acts as a dielectric material, which is equivalent to a friction layer of conventional triboelectric nanogenerator. When the electrification liquid flows through the surface of silicone, the triboelectric effect appears, leading to the electrification liquid and the surface of the silicone are charged with opposite charges, respectively. This is the first step of electricity generating process of the BSNG.

We found that the output performances of deionized water as electrification liquid in liquid-solid contact triboelectrification mode behave better than other fluent materials such as ion solution and organic solvent, according to the related fundamental research works about liquid-solid contact triboelectrification:

[4] Water-Solid Surface Contact Electrification and its Use for Harvesting Liquid-Wave Energy [J]. *Angewandte Chemie*, 2013, 125(48): 12777-12781.

[5] Self-Powered Ion Concentration Sensor with Triboelectricity from Liquid-Solid Contact Electrification [J]. *Advanced Electronic Materials*, 2016, 2(5): 1600006.

Furthermore, the purer the deionized water, the better the output performance. This might be explained by that higher dielectric constant of the electrification liquid, more charges would be generated in the triboelectrification process.

Considering that the BSNG is a wearable device, the biosafety of materials used in it needs to be guaranteed. At the same time, the source of the material and the difficulty of preparation are also important factors to be taken into consideration. From the above, the advantages of high dielectric constant, good bio-safety, low cost and easy

preparation make deionized water an ideal electrification liquid material for BSNG.

Revise in manuscript:

(Page:4. Line:81-83)

The electrification liquid used in BSNG is deionized water, with the advantages of high dielectric constant, good bio-safety, low cost and easy preparation.

Comment 3:

In figure 2c, it is necessary to express the display of the charge in a little more detail. The number of electrons when the electrification liquid enters the bionic channel and the amount of charge in the ionic solution electrode are not proportional to the figure.

Responses:

We are so grateful for your insightful advice. According to your suggestions, we have optimized the model of electron transfer when BSNG working, which is easier for readers to understand. We have modified the Fig. 2c in the manuscript, including that adding the number of the electrons for display and correcting the charge distribution in each stage of a working cycle for BSNG.

Revise in manuscript:

(Page:24. Line:531-531)

Fig. 2 | Working principle and stretchability of BSNG. (c) Schematic diagram of the working mechanism of BSNG.

Comment 4:

In figure 3d, is there any reason to represent the voltage value as a negative value? Looking at figures e and f, short-circuit current and short-circuit charge are expressed as positive values, while open-circuit voltage is expressed as negative values.

Responses:

Thank you for your time and attention to our manuscript. In this work, the open-circuit voltage, short-circuit current and short-circuit charge of BSNG were measured by a Keithley 6517 electrometer, and the data were collected and recorded by an oscilloscope (LeCroy HDO6104). Similarly, there are some research works have also measured the opposite values of the open-circuit voltage and the short-circuit charge of triboelectric nanogenerator by same types of measurements, such as:

[6] A highly shape-adaptive, stretchable design based on conductive liquid for energy harvesting and self-powered biomechanical monitoring [J]. *Science advances*, 2016, 2(6): e1501624.

“The voltage, charge, and current of the saTENGs were measured by a programmable electrometer (part no. 6514, Keithley), and the data were collected and recorded by

computer-controlled measurement software written in LabVIEW.” As shown in Figure R2.

Figure R2. The open-circuit voltage (V_{oc}) and short-circuit transferred charge (Q_{sc}) of the attached-electrode contact-mode saTENG with (A) forward connection and (B) reverse connection to the liquid electrode.

[7] Ultrastretchable, transparent triboelectric nanogenerator as electronic skin for biomechanical energy harvesting and tactile sensing [J]. *Science advances*, 2017, 3(5): e1700015.

“The voltage and charge quantity were recorded by a Keithley electrometer 6514, and the current was recorded with a Stanford low-noise preamplifier SR570.” As shown in Figure R3.

Figure R3. (A) Open circuit voltage V_{oc} (B) short-circuit charge quantity Q_{sc} , and (C) short-circuit current I_{sc} of a PDMS-STENG.

Meanwhile, we try to give a possible explanation for this phenomenon, which could be interpreted by the measurements of the open-circuit voltage, short-circuit current and short-circuit charge of TENG through an electrometer. The value of charge we measured is the amount of transferred charge between the two electrodes, which is a directional value, as shown in Figure R4.

Figure R4. Schematic diagram of the electrical measurement of TENG.

In a model of TENG, when two materials contact and separate from each other, the surface of the two materials will carry an equal amount of opposite charges due to the triboelectric effect, and form a potential difference between the two electrodes due to electrostatic induction. According to a capacitance model (Equation 1) and charge is conserved (Equation 2):

$$Q_R = C V \quad (1)$$

$$Q_R + Q_T = 0 \quad (2)$$

$$Q_T = -C V \quad (3)$$

$$I = \frac{dQ_T}{dt} = -\frac{dCV}{dt} \quad (4)$$

Here, Q_R refers to the amount of charge retained on the surface of the material; C refers to the inherent capacitance of TENG; Q_T refers to the transferred charge between the two electrodes of TENG, which is the short-circuit charge we measured actually; V is open-circuit voltage and I is short-circuit current of TENG. Because the

charge is conserved, the value of Q_T is opposite to the value of Q_R (Equation 2), which leads to the short-circuit charge is opposite to the value of V (Equation 3). The I is the integral of the transferred charge over time (Equation 4), which is also opposite to the value of the V . Therefore, whether the connection of the test instrument and TENG is forward or reversed, the measured value of the open-circuit voltage is always opposite to the value of short-circuit current and short-circuit charge.

In order to facilitate statistics and analysis of the output performance of triboelectric nanogenerator, there are also many studies that unify the direction of the open-circuit voltage, short-circuit current and short-circuit charge of triboelectric nanogenerator, such as:

[8] Vitrimer Elastomer-Based Jigsaw Puzzle-Like Healable Triboelectric Nanogenerator for Self-Powered Wearable Electronics [J]. *Advanced Materials*, 2018, 30(14), 1705918.

“The open-circuit voltage (V_{oc}), short-circuit transferred charge (Q_{sc}) and short-circuit current (I_{sc}) were measured by a Keithley Instruments 6514 electrometer (Solon, OH, USA) and recorded with homemade LabVIEW program.” As shown in Figure R5.

Figure R5. (A) Open circuit voltage V_{oc} (B), Short-circuit current I_{sc} , and (C) short-circuit transferred charge Q_{sc} of the VTENG.

In our work, we prefer to show the original test data rather than unify the direction of the open-circuit voltage, short-circuit current and short-circuit charge of TENG. It should be noted that the Keithley 6517 electrometer has the same function of Keithley 6514 electrometer.

Revise in manuscript:

(Page:17. Line:360-362)

The open-circuit voltage, short-circuit current and short-circuit charge of BSNG were measured by a Keithley 6517 electrometer, and the data were collected and recorded by an oscilloscope (LeCroy HDO6104).

Comment 5:

In figure 4d, we think there will be more important factors that can show differences in each posture. For example, the time difference between each cycle needs to be analyzed.

Responses:

We are so grateful for your instructive advice. We have further analyzed the data of the recorded motion signals of each swimming stroke, and extracted the average motion amplitude and average time interval of the peaks of the motion signals of each stroke. From the statistical results shown in Supplementary Fig. 13, we can know that the amplitudes of the leg movements are larger than the arm movements in breaststroke and backstroke, while the amplitudes of the arm movements are larger than the leg movements in the freestyle. When treading water, the arms have almost no movement and only the signals of legs' movements were caught.

The average peak intervals of the motion signals of the legs and arms in each stroke are similar, so the movement frequency of the legs and arms is almost consistent in each stroke, indicating that the arms and legs of the subjects are coordinated in each stroke. The amplitude sorting of the leg movements with four strokes from high to low is: backstroke, breaststroke, tread water, and freestyle. In contrast, the difference in the amplitudes of the arm movements is not obvious in these three strokes (except for treading water).

The time difference of each cycle is different for each stroke, indicating the frequencies of each movement in different strokes are not the same. The movement frequencies of freestyle and tread water are fast and very similar, followed by breaststroke, and backstroke is the slowest. Because in each cycle, the swimmer should complete a movement with higher speed in freestyle and tread water than that in breaststroke. The time required to complete a movement in backstroke is the longest.

With the information acquired from BSNGs, we can analyze the specific case of each movement of the swimmer to estimate the physiology state under the water. For example, when the swimmer's physical strength begins to decrease, exhibiting a fatigue state with the amplitude of the motion signal decreasing and the time interval of each cycle increasing.

Revise in manuscript:

(Page:11. Line:237-242)

The data of the recorded motion signals of each swimming stroke have been further analyzed, the average motion amplitude and average time interval of the peaks of the motion signals of each stroke have been extracted respectively (Supplementary Fig. 13). With the information acquired from BSNGs, we can analyze the specific case of each movement of the swimmer to estimate the physiology state under the water.

Revise in supplementary information:

(Page:17. Line:235-243)

Supplementary Fig. 13. Statistical analysis of motion signals in four strokes. (a-d) Average motion amplitude of the motion signals of four strokes. (e-h) Average time

interval of the peaks of the motion signals of four strokes. (i) Statistical average motion amplitude of the motion signals of four strokes. (j) Statistical average time interval of the peaks of the motion signals of four strokes. (k) Average motion amplitude of different body parts in four strokes. (l) Average time interval of different body parts in four strokes.

(Page:17-19. Line:244-269)

The data of the recorded motion signals of each swimming stroke have been further analyzed, and the average motion amplitude and average time interval of the peaks of the motion signals of each stroke have been extracted respectively (Supplementary Fig. 13). From the statistical results, we can know that the amplitudes of the leg movements are larger than the arm movements in breaststroke and backstroke, while the amplitudes of the arm movements are larger than the leg movements in the freestyle. When treading water, the arms have almost no movement and only the signals of legs' movements are caught.

The average peak intervals of the motion signals of the legs and arms in each stroke are similar, so the movement frequency of the legs and arms is almost consistent in each stroke, indicating that the arms and legs of the subjects are coordinated in each stroke. The amplitude sorting of the leg movements with four strokes from high to low is: backstroke, breaststroke, tread water, and freestyle. In contrast, the difference in the amplitudes of the arm movements is not obvious in these three strokes (except for treading water).

The time difference of each cycle is different for each stroke, indicating the frequencies of each movement in different strokes are not the same. The movement frequencies of freestyle and tread water are fast and very similar, followed by breaststroke, and backstroke is the slowest. Because in each cycle, the swimmer should complete a movement with higher speed in freestyle and tread water than that in breaststroke. The time required to complete a movement in backstroke is the longest. With the information acquired from BSNGs, we can analyze the specific case of each movement of the swimmer to estimate the physiology state under the water. For

example, when the swimmer's physical strength begins to decrease, exhibiting a fatigue state with the amplitude of the motion signal decreasing and the time interval of each cycle increasing.

Comment 6:

In figure 5, regarding the Undersea rescue system, you said that the wireless transmission and reception were done by charging the capacitor for 4 hours and 30 minutes. I wonder that the person must swim without resting for 4 hours and 30 minutes to charge the capacitor, and I wonder how the connection between wireless transmitter and the rectifier is made. We think you can improve applications using BSNG, is there other examples that can expand application of BSNG?

Responses:

Thanks for your insightful and professional advice. For undersea rescue system, in order to ensure an effective transmission distance, ample energy is required. We choose a 100 μF capacitor to store the energy generated by the BSNG and drive the wireless transmitter. For characterizing the output performance of BSNG, a 100 μF capacitor was charged from 0 V to 3 V by BSNG in about 4.5 hours when tested in lab, and then we use it to drive a wireless transmitter to emit a trigger signal.

According to the charging curve of BSNG and energy consumption of wireless transmission tested in our lab, we estimated that the energy scavenging from about 4 hours' activities of swimming can be sufficient to drive the wireless transmitter to emit a trigger signal.

As we mentioned in manuscript “the 100 μF capacitor charged by four rectified BSNGs wore on human body for about 4 hours”, the total duration of 4 hours is the whole time of experimenter wearing the diving suit, includes on land activities, swimming and taking breaks. When reaching the estimated time, the experimenter performed a demonstration of simulating drowning and launched a rescue signal to power the rescue warning light successfully.

In fact, the operation threshold voltage of the wireless transmitter is 2.2 V. In practical application, it usually does not need to start charging an energy storage device from 0

V. The energy storage device should be precharged. In this case, the BSNG just need to charge a capacitor upon the operation threshold voltage for launching wireless signal. In our experiment, when a trigger signal is transmitted, the voltage of the capacitor will drop from 2.2 to about 0.9 V. We recharge this 100 μF capacitor from 0.9 V to 2.2 V within 1 hour by the BSNG in our lab.

Revise in supplementary information:

(Page:20. Line:277-279)

Supplementary Fig. 15. Recharging a 100 μF capacitor by BSNG after the transmitter launching a wireless signal.

(Page:20. Line:281-287)

The operation threshold voltage of the wireless transmitter is 2.2 V. In practical application, it usually does not need to start charging an energy storage device from 0 V. The energy storage device should be precharged. In this case, the BSNG just need to charge a capacitor upon the operation threshold voltage for launching wireless signal. In our experiment, when a trigger signal is transmitted, the voltage of the capacitor will drop from 2.2 to about 0.9 V. We recharge this 100 μF capacitor from 0.9 V to 2.2 V within 1 hour by the BSNG in our lab.

The connection between wireless transmitter and the rectifier is shown in Fig. 5b and

d. A 100 μF capacitor and four rectifiers were all welding on the wireless transmitter. For ease of understanding, we have modified some of the marks and presentations in Fig. 5b and d of the manuscript.

Revise in manuscript:

(Page:27. Line:557-557)

Fig. 5 | Undersea rescue system based on BSNG. (b) Integrated wireless transmitter. (d) Simple circuit diagram of undersea rescue system.

We are so grateful for your instructive suggestions about “improve applications using BSNG”. We also think that the applications of BSNG should be improved to play a greater role in the field of self-powered electronics. In this work, we primarily aimed at proving the feasibility of using BSNG as an underwater energy harvesting and motion sensing, which is the unique application scenario of our BSNG.

According to your suggestions, we further presented some typical applications of the BSNG (Supplementary Fig. 16). The BSNG can also be applied in other application scenarios: The first application of BSNG is as a controller. By coupling with a traditional FET, BSNG can be used for active modulation of conventional electronics, for example, to control the blinking of an LED through BSNG. Using the electrostatic potential created by triboelectrification as a “gate” voltage to tune/control electrical transport and transformation in semiconductors, BSNG has established a direct

interaction mechanism between the human and electronics. In the near future, BSNG may also be used for electronic skin, human-machine interface, etc.

Another application of BSNG is on land energy harvesting. The energy harvested by a wearable BSNG can be stored and drive some portable appliances, such as healthcare monitoring, humidity sensor, temperature sensor, LCD display, etc. The third application of BSNG is acted as a wearable sensor for athletic exercise. For example, the BSNG can be used for some common movement monitoring in daily life, such as body-building and marathon.

Revise in manuscript:

(Page:12. Line:263-265)

It is noted that BSNG also has a wealth of applications in daily life (Supplementary Fig. 16), such as harvesting energy or self-powered monitoring when body-building, establishing human-machine interface, and so on.

Revise in supplementary information:

(Page:21. Line:288-292)

Supplementary Fig. 16. Three types of applications for BSNG. (a-c) BSNG used as a controller and the outlook for BSNG used as electronic skin. (d-f) BSNG used as an energy harvester to drive some low-power appliances. (g-i) BSNG used as self-powered wearable devices for monitoring motions in human's daily life.

(Page:21-22. Line:294-307)

The BSNG can also be applied in other application scenarios: The first application of BSNG is as a controller. By coupling with a traditional FET, BSNG can be used for active modulation of conventional electronics, for example, to control the blinking of an LED through BSNG. Using the electrostatic potential created by triboelectrification as a “gate” voltage to tune/control electrical transport and transformation in semiconductors, BSNG has established a direct interaction mechanism between the human and electronics. In the near future, BSNG may also be used for electronic skin, human-machine interface, etc.

Another application of BSNG is on land energy harvesting. The energy harvested by

a wearable BSNG can be stored and drive some portable appliances, such as healthcare monitoring, humidity sensor, temperature sensor, LCD display, etc. The third application of BSNG is acted as a wearable sensor for athletic exercise. For example, the BSNG can be used for some common movement monitoring in daily life, such as body-building and marathon.

Comment 7:

In Supplementary Figure 1a, it needs more information about “CTE”

Responses:

We are so grateful for your instructive advice. We have supplemented the annotation in Supplementary Fig. 1 (a), and we have added information about the coefficient of thermal expansion (CTE) in the description of Supplementary Fig. 1.

Thermal expansion is the tendency of matter to change its shape, area, and volume in response to a change of temperature. Temperature is a monotonic function of the average molecular kinetic energy of a substance. When a substance is heated, the kinetic energy of its molecules increases. Thus, the molecules begin vibrating/moving more and usually maintain a greater average separation. So most materials will expand as the temperature increases. The relative expansion divided by the change in temperature is called the material's coefficient of thermal expansion (CTE) and generally varies with temperature. The coefficient of thermal expansion describes how the size of an object changes with a change in temperature. Specifically, it measures the fractional change in size per degree change in temperature at a constant pressure. (Ref. :

[9] *Physics for Scientists and Engineers* - Volume 1 Mechanics/Oscillations and Waves/Thermodynamics. New York, NY: Worth Publishers. (2008) pp. 666–670. ISBN 978-1-4292-0132-2.

[10] Giant thermal expansion and α -precipitation pathways in Ti-alloys[J]. *Nature communications*, 2017, 8(1): 1429.)

Revise in supplementary information:

(Page:4. Line:54-54)

Supplementary Fig. 1. The phenomenon of stress mismatch between silicone and PDMS. (a) The mechanism of stress mismatch between silicone and PDMS.

(Page:5. Line:69-78)

Thermal expansion is the tendency of matter to change its shape, area and volume, in response to a change of temperature. Temperature is a monotonic function of the average molecular kinetic energy of a substance. When a substance is heated, the kinetic energy of its molecules increases. Thus, the molecules begin vibrating/moving more and usually maintain a greater average separation. So most materials will expand as the temperature increases. The relative expansion divided by the change in temperature is called the material's coefficient of thermal expansion (CTE) and generally varies with temperature. The coefficient of thermal expansion describes how the size of an object changes with a change in temperature. Specifically, it measures the fractional change in size per degree change in temperature at a constant pressure^{3,4}. (Ref. :

[3] Tipler, P. A. & Mosca, G. Physics for scientists and engineers. (Macmillan, 2007).

[4] Bönisch, M. et al. Giant thermal expansion and α -precipitation pathways in Ti-alloys. Nat Commun 8, 1429 (2017).

Comment 8:

In figure 2e, it needs to be uniformity of spacing, “Strain =0% > Strain=0%”

Responses:

Thank you for your time and attention to our manuscript. We have modified the annotation in Fig. 2e of the manuscript.

Revise in manuscript:

(Page:24. Line:531-531)

Fig. 2 | Working principle and stretchability of BSNG. (e) BSNG (indicated by red frame) at initial state (0 % strain) and stretched state (60 % strain).

Comment 9:

In figure 5e, the expression "simple circuit diagram" is inappropriate.

Responses:

Thank you for your time and attention to our manuscript. We have modified the legend of Fig. 5 in the manuscript.

Revise in manuscript:

(Page:27. Line:555-561)

Fig. 5 | Undersea rescue system based on BSNG. Photograph of undersea rescue system which included (a) integrated energy harvesting diving suit, (b) integrated wireless transmitter and (c) wireless receiver integrated with a red warning light. (d) Simple circuit diagram of undersea rescue system. (e) Voltage changes of a 100 μ F capacitor charged by BSNG and used to power a wireless transmitter to emit a trigger signal. (f) Physical map of undersea rescue system sending an alert when swimmer in danger (red LED was lighted up remotely).

Comment 10:

In figure 5f, what is the white pole that the subject's right hand is holding?

Responses:

Thank you for your time and attention to our manuscript. The white pole holding in the subject's right hand is a pole for rescue in the swimming pool used by staffs. We use it to demonstrate the process of simulating underwater rescue. We have already marked it in Fig. 5f of the manuscript.

Revise in manuscript:

(Page:27. Line:557-557)

Fig. 5 | Undersea rescue system based on BSNG. (f) Physical map of undersea rescue system sending an alert when swimmer in danger (red LED was lighted up remotely).

Reviewer #2 (Remarks to the Author):

In this paper, authors introduced an interesting bionic stretchable nanogenerator (BSNG) inspired by the structure of ion channels on the cytomembrane of electrocyte. The device is with two different electricity generation modes, great stretchability and superior output performance under water, which is of great value to wearable electronic devices for underwater application. The human body multi-position motion monitoring and undersea rescue system based on the BSNG was demonstrated in this manuscript. They showed an outstanding tensile fatigue resistance over 50,000 times for the good opportunity as a soft, wearable, sustainable power sources used for underwater electronics. It is definitely a novel device of its unique design and interdiscipline of nanogenerator, bionics and biomedical engineering, which is first presented in research works. The paper merits publication after a few minor changes.

Comment 1:

The macro size of the entire BSNG is actually not small, why is it still called a nanogenerator?

Responses:

Thank you for your time and attention to our manuscript. In 2006, the first ZnO nanowire-based nanogenerator (NG) was proposed, which utilizes piezoelectric effect of nanowires for converting tiny mechanical energy into electricity. This research has opened up the field of nano energy. Since then, various nanogenerators have been demonstrated to harvest the micro/nano energy from our living environment. In 2012, the first triboelectric nanogenerator (TENG) was proposed, which was based on the conjunction of triboelectrification effect and electrostatic induction. Using the electrostatic charges created on the surfaces of two dissimilar materials when they are brought into physical contact, which can generate a potential drop when the two surfaces are separated by a mechanical force. Then electrons will be drove to flow between the two electrodes built on the top and bottom surfaces of the two materials. The fundamental theory of the nanogenerators is starting from the Maxwell equations, and the second term in the Maxwell's displacement current is directly related to the

output electric current of the nanogenerators. The nanogenerators are the applications of Maxwell's displacement current in energy and sensors, and the Maxwell's displacement current is the origin of nanogenerators. General speaking, the energy conversion devices that use piezoelectric effect or triboelectrification effect and electrostatic induction, based on Maxwell's displacement current, can be called nanogenerators. In this paper, the proposed BSNG is also based on the method of Maxwell's displacement current, which utilizing triboelectrification effect and electrostatic induction between liquid and soft materials. Therefore, although the macro size of BSNG is not small, we still name it bionic stretchable nanogenerator. (Ref:

[1] Piezoelectric nanogenerators based on zinc oxide nanowire arrays[J]. *Science*, 2006, 312(5771): 242-246.

[2] Flexible triboelectric generator[J]. *Nano energy*, 2012, 1(2): 328-334.

[3] On Maxwell's displacement current for energy and sensors: the origin of nanogenerators[J]. *Materials Today*, 2017, 20(2): 74-82.)

Comment 2:

Why do authors choose deionized water as the electrification liquid in BSNG instead of other liquids? What are the advantages of deionized water compared to other liquids?

Responses:

Thank you for your time and attention to our manuscript. The electrification liquid in BSNG acts as a dielectric material, it corresponds to a friction layer of conventional triboelectric nanogenerator. When the electrification liquid flows through the surface of silicone, the triboelectrification effect generated, led to the electrification liquid and the surface of the silicone are respectively charged with opposite charges. This is the first step in generating electricity from BSNG. Therefore, the electrification liquid first needs a relatively high dielectric constant. According to the related research works of triboelectric nanogenerator based on liquid-solid contact electrification, we found that compared with other fluent materials such as ion solution and organic solvent, the deionized water always behave best output performance as electrification liquid in liquid-solid contact triboelectrification mode. And the purer the water, the better the

output performance. In addition, as a wearable electronic device, the biosafety of the materials used needs to be guaranteed. Compared to other ion solutions and organic solvents, deionized water is with good biosecurity. At the same time, the source of the material and the difficulty of preparation are also important factors to consider. From the above, good output performance, bio-safety, low cost and easy preparation make deionized water an ideal electrification liquid material. (Ref. :

[4] Water-Solid Surface Contact Electrification and its Use for Harvesting Liquid-Wave Energy [J]. *Angewandte Chemie*, 2013, 125(48): 12777-12781.

[5] Self-Powered Ion Concentration Sensor with Triboelectricity from Liquid–Solid Contact Electrification [J]. *Advanced Electronic Materials*, 2016, 2(5): 1600006.)

Comment 3:

The conductivity of sodium chloride solution is not excellent compared to other commonly used conductive ion electrode such as lithium chloride solution. Does the conductivity of the electrode influence the performance of the nanogenerator based device? Why to choose sodium chloride solution here?

Responses:

Thank you for your time and attention to our manuscript. The inherent impedance of triboelectric nanogenerator (TENG) is typically hundreds of megohms, whereas the resistance of the ion solution electrode (diameter, 10 mm; length, 50 mm) is usually range from the order of several thousand ohms to tens of kilohms, which is much smaller than the inherent impedance of TENG. For this reason, the high inherent impedance of TENG allows the resistance of its electrode to vary within a wide range without much degradation of the performance.

TENG is based on the electrostatic induction and has inherent capacitive behavior. Its intrinsic impedance can be estimated to be:

$$Z = \frac{1}{2\pi fC}$$

Where, f is frequency of external force; C is inherent capacitance between single electrode and ground for single-electrode-mode TENG, or average capacitance between

two electrodes for two-electrodes-mode TENG. For a rationally designed TENG, the two electrodes are fully-insulated and the internal resistance is nearly infinity. Since TENG's inherent capacitance is generally in the order of tens or hundreds of pF; if the external force has a low frequency (about 0.4 to 4 Hz in the experiments), the intrinsic impedance of the TENG will be very high (in the range of hundreds of MΩ).

For a TENG that has a negligible electrode resistance, when it is connected to a resistive load, the equivalent circuit model can be shown as:

If we consider the effect of the TENG's electrode resistance, the equivalent circuit model now adds a new term, which is the electrode resistance r .

The power applied to the external resistance R can be estimated by: (For simplification, the V_{oc} is assumed as a sinusoidal profile in the analysis.)

$$P = \frac{V_{oc}^2 R}{\frac{1}{\omega^2 C^2} + (r + R)^2}$$

It is clearly shown from the above equation that when r is smaller than $1/100 \omega C$, the influence of r on P is very small. Since TENG's inherent impedance is generally in the range of hundreds of MΩ, the performance of the TENG will change very slightly as long as the TENG's electrode resistance is smaller than the MΩ range.

Therefore, the conductivity of sodium chloride solution is sufficient for the electrode of BSNG. Besides that, the good biosafety, wide range of sources and simple

preparation method of sodium chloride solution are also important reasons for choosing it as an electrode. (Ref:

[6] Ultrastretchable, transparent triboelectric nanogenerator as electronic skin for biomechanical energy harvesting and tactile sensing[J]. *Science advances*, 2017, 3(5): e1700015.

[7] A highly shape-adaptive, stretchable design based on conductive liquid for energy harvesting and self-powered biomechanical monitoring[J]. *Science advances*, 2016, 2(6): e1501624.)

Comment 4:

How to prove that bamboo joint-liked microstructures at the bottom of the channels can make the channels more hydrophobic? Please add a pro-hydrophobic test of the channels with and without bamboo joint-liked microstructures.

Responses:

We are so grateful for your instructive advice. We have added the pro-hydrophobic test of the channels with and without bamboo joint-liked microstructures. The results (Supplementary Fig. 4) show that the deionized water has a larger contact angle with the surface of silicone with bamboo joint-liked microstructures compared to the surface of silicone without microstructures, which demonstrate that the surface of the silicone with microstructure has better hydrophobicity than the smooth surface without microstructure.

Revise in supplementary information:

(Page:8. Line:101-103)

Supplementary Fig. 4. Contact angle test results of the silicone with smooth surface (a) and the silicone with bamboo joint-like microstructure surface (b).

(Page:8. Line:104-108)

The Contact angle test results (Supplementary Fig. 4) show that the deionized water has a larger contact angle with the surface of silicone with bamboo joint-like microstructures compared to the surface of silicone without microstructures, which demonstrate that the surface of the silicone with microstructure has better hydrophobicity than the smooth surface without microstructure.

Comment 5:

What is the thickness of the electrification layer and induction layer of BSNG? Please specify the size of each part structure of BSNG in the manuscript, these data are important for this device.

Responses:

We are so grateful for your advice. The thickness of the electrification layer and induction layer of BSNG is 2 mm and 1 mm respectively. For the sake of understanding, we use three views of BSNG (Supplementary Fig. 5) to help illuminate:

The whole size of the BSNG is 6 cm × 10 cm × 8 mm. For the electrification layer of BSNG, the size of the area of bionic channels is 2 cm × 8 cm × 2 mm and the size of the fluid chamber is 1 cm × 8 cm × 2 mm. The size of the connecting pipelines between bionic channels and fluid chamber is 1 cm in length and 1.5 mm in diameter. For the induction layer of BSNG, the sizes of two electrodes are 2 cm ×

8 cm × 1 mm and 1 cm × 8 cm × 1 mm, facing to the region of bionic channels and fluid chamber of the electrification layer respectively. The thickness of the silicone layer between the electrification layer and the induction layer is 1 mm. The thickness of the silicone layer wrapped around the electrification layer and the induction layer is 2 mm.

Revise in manuscript:

(Page:5. Line:96-97)

The detailed size of each part structure of BSNG can be found in Supplementary Fig. 5.

Revise in supplementary information:

(Page:8. Line:109-111)

Supplementary Fig. 5. Three views of BSNG and the specific dimensions of each part structure of BSNG.

(Page:9. Line:112-120)

The whole size of the BSNG is 6 cm × 10 cm × 8 mm. For the electrification layer of BSNG, the size of the area of bionic channels is 2 cm × 8 cm × 2 mm, the

size of the fluid chamber is 1 cm × 8 cm × 2 mm. The size of the connecting pipelines between bionic channels and fluid chamber is 1 cm in length and 1.5 mm in diameter. For the induction layer of BSNG, the sizes of two electrodes are 2 cm × 8 cm × 1 mm and 1 cm × 8 cm × 1 mm, facing to the region of bionic channels and fluid chamber of the electrification layer respectively. The thickness of the silicone layer between the electrification layer and the induction layer is 1 mm. The thickness of the silicone layer wrapped around the electrification layer and the induction layer is 2 mm.

Comment 6:

For a stretchable device, 50 thousand times of fatigue test results of BSNG are excellent. However, after 50 thousand times uniaxial tensile test, will the internal structure of BSNG has some changes? The reviewer suggests to add more details (e.g. photograph) to show the conditions of the internal structure of BSNG after uniaxial tensile test.

Responses:

We appreciate the reviewer very much for this significant comment. After 50 thousand times uniaxial tensile test, the output performance of BSNG is not attenuated, and BSNG has a complete structure without any damage from the outside. To prove that the internal structure of BSNG also has no changes, we cut open the electrode layer on the back of the BSNG to observe whether the structure has changed. From the optical photos (Supplementary Fig. 12a and b), the internal channel structure of BSNG is intact without any damage (Red ink is injected to see the internal structure of BSNG more clearly). In addition, we have also characterized the internal bionic channel structure by SEM (Supplementary Fig. 12c and d). From the SEM photos of bionic channels of BSNG, we can see that the bamboo joint-linked microstructures at the bottom of the channels is also intact. Benefiting from the small friction stress between liquid and soft matter, the bionic channels inside the BSNG is not easily damaged even after a long period of reciprocating flow of the electrification liquid.

Revise in manuscript:

(Page:10. Line:209-214)

The characterization of the internal structure of BSNG after 50 thousand times fatigue test demonstrates that the bionic channels inside BSNG is intact (Supplementary Fig. 12). Benefiting from the small friction stress between liquid and soft matter, the bionic channels inside BSNG is not easily damaged even after a long period of reciprocating flow of the electrification liquid.

Revise in supplementary information:

(Page:16. Line:218-221)

Supplementary Fig. 12. Characterization of the internal structure of BSNG after 50,000 times fatigue test. (a-b) Optical photos of the internal structure of BSNG, (c-d) SEM photos of the bionic channels of BSNG.

(Page:16-17. Line:222-233)

After 50 thousand times uniaxial tensile test, the output performance of BSNG is not attenuated, and BSNG has a complete structure without any damage from the outside.

To prove that the internal structure of BSNG also has no changes, the electrode layer on the back of the BSNG was cut open to observe whether the structure has changed. From the optical photos (Supplementary Fig. 12a and b), the internal channel structure of BSNG is intact without any damage (Red ink is injected to see the internal structure of BSNG more clearly). In addition, the internal bionic channel structure was also characterized by SEM (Supplementary Fig. 12c and d). From the SEM photos of bionic channels of BSNG, the bamboo joint-like microstructures at the bottom of the channels is also intact. Benefiting from the small friction stress between liquid and soft matter, the bionic channels inside the BSNG is not easily damaged even after a long period of reciprocating flow of the electrification liquid.

Comment 7:

The power density curve is also an important criterion for the BSNG. Please add relevant experimental data.

Responses:

We are so grateful for your instructive advice. We supplemented the power density curves of BSNG in two different working modes. The power density curves of BSNG in single electrode mode and liquid-solid contact mode are shown in Supplementary Fig. 11a-b and c-d respectively. In single electrode mode, the peak power density of BSNG can reach 18 mW/m^2 when the external load is $50 \text{ M}\Omega$. In liquid-solid contact mode, the peak power density of BSNG can reach $62.5 \text{ }\mu\text{W/m}^2$ when the external load is $300 \text{ M}\Omega$.

Revise in manuscript:

(Page:9. Line:194-199)

The power density curves of BSNG in single electrode mode and liquid-solid contact mode are shown in Supplementary Fig. 11 respectively. In single electrode mode, the peak power density of BSNG can reach 18 mW/m^2 when the external load is $50 \text{ M}\Omega$. In liquid-solid contact mode, the peak power density of BSNG can reach $62.5 \text{ }\mu\text{W/m}^2$ when the external load is $300 \text{ M}\Omega$.

Revise in supplementary information:

(Page:15. Line:211-217)

Supplementary Fig. 11. Power density curves of BSNG in two different working modes. (a) Open-circuit voltage and short-circuit current for single electrode mode of BSNG at different load resistances. (b) Peak power density for single electrode mode of BSNG at different load resistances. (c) Open-circuit voltage and short-circuit current for liquid-solid contact mode of BSNG at different load resistances. (d) Peak power density for liquid-solid contact mode of BSNG at different load resistances.

Comment 8:

What are the challenges of BSNG, and what improvements can be made afterwards?

The authors should discuss more in the “Discussion” section.

Responses:

We are so grateful for your insightful advice. We have added some discussion and references about the challenges and the outlook of BSNG in “Discussion” section of

the manuscript according to your suggestion. Even though the BSNG in this work is flexible, stretchable and behaving well underwater, there is still much room for improvement. The dimensions of BSNG could be smaller and thinner if micromachining can be applied to the preparation of BSNG. A smaller and thinner BSNG may be a potential body mechanical energy harvester for implantable applications. Furthermore, some packaging and insulating treatments could be used to further enhance performance of BSNG underwater by reducing electrostatic shielding of water. Materials optimization and surface modifications could also be explored to improve the output performance of BSNG by maximizing the surface electrostatic charge density while liquid flowing. In addition, it is also possible to design the BSNG into arbitrary, complicated shapes by changing different structures of the channel. These characters are essentials for soft wearable devices power source. With these improvements, BSNG could have opened up much more opportunities for many potential applications ranging from electrical skins, soft robots, wearable electronics, to implantable medical devices.

Revise in manuscript:

(Page:13-14. Line:284-294)

Even though the BSNG in this work is flexible, stretchable and behaving well underwater, there is still much room for improvements. The dimensions of BSNG could be smaller and thinner in virtue of developments of micromachining techniques in the future. In addition, some advances of materials, surface modifications and packaging strategies could be used to further enhance performance of BSNG by maximizing surface electrostatic charge density while liquid flowing and reducing electrostatic shielding of ions underwater. Meanwhile, a miniaturized BSNG has a potential to act as body mechanical energy harvester or sensor for implantable applications, such as harvesting heart beating energy and sensing pulse signals. With these improvements, BSNG could have opened up much more opportunities for many potential applications of electrical skins, soft robots, wearable electronics and implantable medical devices.

Comment 9:

There are some basic language issues in the manuscript, the authors should check carefully and modify them.

Responses:

Thank you for your time and attention to our manuscript. We have carefully examined the entire manuscript, and modified the grammar and spelling errors. Such as:

(Page:17. Line:367)

The phenomenon of stress mismatch between silicone and PDMS. > Phenomenon of stress mismatch between silicone and PDMS.

(Page:17. Line:369)

The mechanosensitive multiple channels of BSNG. > Mechanosensitive multiple channels of BSNG.

(Page:17. Line:373)

The progress of Na⁺ and K⁺ transfer in cell membrane. > Progress of Na⁺ and K⁺ transfer in cell membrane.

(Page:18. Line:387)

The fully fabrication process of the BSNG. > Fully fabrication process of the BSNG.

REVIEWERS' COMMENTS:

Reviewer #1 (Remarks to the Author):

I think the authors conduct well additional experiments and correct some sentences that can cause the readers confusing.

It would be acceptable in Nature Communications, high-quality journal.

Reviewer #2 (Remarks to the Author):

The authors carefully addressed the reviewers' comments and the manuscript is further enhanced after the revision. The research topic and the new patch for powering the soft wearable electronics used underwater by nanogenerators are very interesting. Thus, it is recommended to the acceptance.